# The Effect of Trap Color on Catches of *Monochamus galloprovincialis* and Three Most Numerous Non-Target Insect Species

**DOI:** 10.3390/insects13030220

**Published:** 2022-02-22

**Authors:** Lidia Sukovata, Aleksander Dziuk, Radosław Plewa, Tomasz Jaworski

**Affiliations:** Department of Forest Protection, Forest Research Institute, 3, Braci Leśnej St., Sękocin Stary, 05-090 Raszyn, Poland; a.dziuk@ibles.waw.pl (A.D.); r.plewa@ibles.waw.pl (R.P.); t.jaworski@ibles.waw.pl (T.J.)

**Keywords:** cerambycidae, clerid beetles, color, cross-vane traps, immature and mature longhorned beetles, *Pinus sylvestris*, RAL, reflectance, *Spondylis buprestoides*, *Thanasimus* spp.

## Abstract

**Simple Summary:**

The pine sawyer, *Monochamus galloprovincialis*, is a longhorned beetle widespread in Europe. It develops in severely weakened, dying, or recently dead pine trees. The importance of *M. galloprovincialis* has increased since it was shown to be a vector of the alien and invasive pine wood nematode, *Bursaphelenchus xylophilus*, which can kill pines within a year. Pheromone traps are the most useful tools for monitoring *M. galloprovincialis*. While black traps are most commonly used, the objective of our studies was to test the attractiveness of different colors to immature and mature *M. galloprovincialis* and three non-target species. The results could be useful in selecting an optimal color that is attractive to *M. galloprovincialis*, but minimizes bycatch of non-target insects. A total of twenty colors were tested, including nine colors tested in the field, using cross-vane traps. The unpainted white traps were found to be most attractive to *M. galloprovincialis* and can be used to increase catches of this insect. However, the predatory beetles *Thanasimus* spp. responded to the trap color in the same way as *M. galloprovincialis*; therefore, either trap design or lure composition should be modified to reduce the impact on these beneficial insects.

**Abstract:**

Black pheromone-baited traps are commonly used for monitoring *Monochamus galloprovincialis*, a vector of *Bursaphelenchus xylophilus*, although few studies have been conducted on its response to color (black, white, and clear). The objective of our studies was to evaluate the attractiveness of different colors to *M. galloprovincialis* and non-target species: *Spondylis buprestoides* and predatory *Thanasimus formicarius* and *T. femoralis*. Laboratory tests of fifteen colors against immature and mature *M. galloprovincialis* revealed some differences in their color preference. In two field tests, eight colors of coroplast vanes in cross-vane traps were compared with unpainted white (a reference (RF)). The first test confirmed the laboratory results, i.e., RF was slightly more attractive to *M. galloprovincialis* than pastel yellow, reseda green, and cyan blue, but trap color had no significant effect on any of the insect species studied. In the second test, the attractiveness of RF was highest and significantly different from pure white (for all four species), light blue, and pine green (except *S. buprestoides*). Overall, the unpainted white traps appeared to be most effective in catching *M. galloprovincialis*. *Thanasimus* spp. responded to the colors similarly to *M. galloprovincialis*; therefore, either trap design or lure composition should be modified to reduce their catches.

## 1. Introduction

The pine sawyer, *Monochamus galloprovincialis* (Oliv.) (Coleoptera, Cerambycidae), is a longhorned beetle widespread in Europe. Its main host trees are pines of different species. Females oviposit under the thin bark of weakened, dying, and recently dead pines, but also freshly cut logs. The young larvae develop under the bark, while older larvae bore into the wood, where they pupate. To reach sexual maturity, adults feed on shoots and thin bark of twigs. Generally, *M. galloprovincialis* is not considered a major forest pest. However, its importance has increased since it was shown in Europe to be a vector of the pine wood nematode, *Bursaphelenchus xylophilus* (Steiner and Bührer) Nickle (Rhabditida, Aphelenchoididae) [1], which causes pine wilt disease leading to tree dieback within a year under favorable conditions [2,3,4,5].

Two approaches are recommended for the detection and monitoring of *B. xylophilus*. One is focused on the detection of the nematode in the wood samples collected from symptomatic trees, while the other is based on monitoring its vector and checking the presence of the nematode in beetles [6,7]. Numerous studies have been conducted to develop an optimal trap for catching beetles of *Monochamus* spp., including *M. galloprovincialis*. They focused mainly on the trap type (design), collecting cups (with killing agents or for live insect trapping), and trap treatment with surfactants, e.g., polytetrafluoroethylene (hereafter referred to as PTFE), silicone, etc., to increase the slipperiness of the trap surface [8,9,10,11,12,13,14,15]. However, little attention has been paid to trap color.

For insects, color plays a very important role in host plant/flower selection [16,17,18,19,20,21,22], mate finding [23,24,25,26], predator avoidance [27,28,29], etc. Insects’ visual system is usually trichromatic, and their photoreceptors are maximally sensitive to ultraviolet (UV) (~350 nm), blue (450–480 nm), and green, i.e., long wavelength (LW) (500–550 nm) [30,31]. Although beetles lost their sensitivity to blue wavelengths millions of years ago, some of them were able to overcome this loss through opsin duplications, particularly UV opsins [32,33]. Furthermore, some insects were also found to be sensitive to the violet (~420 nm) and red (>600 nm) regions of the light spectrum [33,34,35]. However, insect response/sensitivity to different wavelengths and reflectance intensity seems to be species- and sex-specific [30,34,36,37,38]. Moreover, the recent study of the emerald ash borer *Agrilus planipennis* Fairmaire (Coleoptera, Buprestidae) showed that immature and mature individuals differ not only in the expression of chemosensory-related genes, but also in the expression of two opsin genes (UV and LW), which may explain the much stronger ability of mature males to visually detect their mates [39].

As in other insects, the type of eyes in longhorned beetles depends on the period of their activity, i.e., during daytime (diurnal) or at night (nocturnal) [40,41], which, in turn, is temperature-dependent. Diurnal insects usually possess apposition eyes, which allow them to see colors, while superposition eyes are characterized by very high absolute sensitivity and the ability to detect light polarization [42]. *Monochamus* spp. beetles are usually active during the day, e.g., *M. saltuarius* Gebler, *M. scutellatus* (Say), and *M. notatus* (Drury) [43]. *M. galloprovincialis* may also be considered a diurnal insect. The only study on the morphological and optical features of eyes in beetles of the genus *Monochamus* was conducted for *M. alternatus* Hope, which is active at night, unlike the above-mentioned species [44]. Surprisingly, its eyes are of the apposition type, such as in diurnal species, but they are adapted to low light intensity [45] and were shown to play an important role in mate finding over short distances [46]. The importance of visual and/or chemical cues in host tree finding was also shown for *M. galloprovincialis*, and its response to relevant stimuli appeared to be dependent on sex and sexual maturity [47].

Color preference has been successfully used to increase the effectiveness of the traps used for monitoring different insect species [48,49,50]. However, for *Monochamus* spp., most studies addressing the effect of trap color on beetle catches have been limited to comparisons of black, white, and/or transparent traps [8,14,15,16,51]. Therefore, the objective of our study was to estimate the response of *M. galloprovincialis* of different sexes and maturity status to a range of colors under laboratory conditions and then to test whether trap color could be used to increase trap effectiveness in the field. In addition, the effect of color on the most commonly captured non-target beetles (*Spondylis buprestoides* (L.) (Cerambycidae) and predatory *Thanasimus formicarius* (L.) and *T. femoralis* (Zetterstedt) (Cleridae)) was assessed, so that this can be taken into account when optimizing a trap for the monitoring of *M. galloprovincialis*.

## 2. Materials and Methods

### 2.1. Testing the Response of Immature and Mature M. galloprovincialis to Different Colors under Laboratory Conditions

In the laboratory tests, we used a concept combining a design of olfactometers and light reflection from a surface painted with different colors. Two glass cylinders, with a length of 30 cm and a diameter of 10.5 cm, were placed linearly at a distance of about 1 cm from each other. The cylinders were connected with cling film and wrapped twice with black agrofiber, leaving 2–3 cm of the outer surfaces uncovered to allow light to reach the inside of the cylinders. Adhesive tape was used to hold the agrofiber in place and strengthen the entire structure. A hole was made in the center for inserting the tested beetles. The structure was placed on the table (Figure 1), parallel to a window. The 15 × 15 cm plates, cut from a white corrugated hollow polypropylene sheet (hereafter referred to as coroplast), were painted with ColorMatic mattish acrylic spray paints of selected colors (Table 1) (paints from the company MIPA SE, Essenbach, Germany, injected in aerosol cans of the ColorMatic series from MotipDupli Group B.V., Wolvega, The Netherlands). The plates of two colors were tested simultaneously. They were placed at the outer holes of the cylinders, at a small angle to the window, in order to allow light reflection from the plates.

The beetles (one per trial) were allowed to make a choice for 3 min, and their response was estimated by observing their presence at the plate of either color. Each pair of colors was tested using 30 beetles, in most cases (Table 2), and the plates were transpositioned after testing half of the beetles in the sample. Tests were performed for immature and mature males and females. Immature beetles were obtained from *M. galloprovincialis*-infested pine tops and thick branches left after stand thinning and collected one year later in the Międzychód Forest District (hereafter FD). To obtain mature beetles, immature individuals were reared for two weeks on pine twigs and shoots with needles in plastic boxes, each sex separately, 10 individuals per box.

Three laboratory tests were performed (Table 2): (1) five pairs of contrasting colors, (2) six pairs of closely related colors, and (3) nine colors compared to the white (unpainted) coroplast. The colors were selected based on the colors present in the environment of *M. galloprovincialis*, e.g., green needles in the crowns, orange thin bark and brown thick bark of Scots pine (a host tree), and blue sky, while white (unpainted) coroplast is the basic material for making vanes for the traps offered for foresters by Chemipan R&D Laboratories (Warsaw, Poland). These colors represent a wide range of wavelengths and light reflectance (Table 1).

### 2.2. Testing the Response of Mature M. galloprovincialis to Different Colors under Field Conditions

The first experiment was conducted in 2019. Four colors, i.e., pastel yellow, cyan blue, reseda green, and white (unpainted) coroplast (Table 1), were selected, based on the laboratory tests (Table 2). A white, unpainted cross-vane IBL-5 trap (Chemipan R&D Laboratories, Poland) was used as a reference trap (Figure 2). Vanes were made of two 50 cm long and 20 cm wide pieces of coroplast and inserted into a 17-cm diameter funnel. The funnel and the lid were made of hard plastic. Trap collectors contained 250–300 mL of 30% water solution of ethylene glycol, with a small amount of an odorless detergent used to reduce liquid surface tension. To obtain the traps of a specific color, the vanes were painted twice with mattish ColorMatic acrylic spray paints (see description in Section 2.1). Then, the vanes and funnels of all traps were treated with a PTFE grease (Boll Trade Agency, Poland). It is important to note that the treated surfaces appeared shiny, due to the presence of mineral oil in the lubricant composition. The traps were baited with Galloprotect Pack lures (SEDQ, Barcelona, Spain), which consist of 2-undecyloxy-1-ethanol (the aggregation pheromone of *Monochamus* spp.), ipsenol and 2-methyl-3-buten-1-ol (the kairomonal chemicals released by bark beetles), and α-pinene (the kairomonal component released by host trees).

The studies were conducted in the Międzychód FD (N 52.6794, E 15.6944), in two 83-year-old pure Scots pine stands with moss vegetation cover. They were selected based on the availability of treetops left on the ground after thinning in the previous year, with signs of *M. galloprovincialis* infestation. Traps were set up in a randomized complete block design across 14 blocks (replicates) of quadratic (2 × 2) shape, with 6 or 4 blocks in each stand, and with each block containing one of each color variant. Traps within and between blocks were separated by approximately 100 m. Traps were set on 24 and 25 June and checked and emptied on 1 and 9 July. They were suspended on metal hooks from dead branches on trees, 4–6 m above the ground, using an adapted telescopic pole. Data from all trap inspections in the same experiment were not treated as replicates.

Captured beetles (*M. galloprovincialis* males and females separately, *S. buprestoides* and *Thanasimus* spp.) were counted in the laboratory, and up to 20 pine sawyer females in each trap were dissected, in order to check for the presence and to count developed eggs, which indicate female maturation. *S. buprestoides* and two *Thanasimus* species were counted only in the samples collected during the first trap inspection.

The second experiment was conducted in 2020. Since no differences were found among the colors tested in 2019 (see Results), we decided to test the following six colors (Table 1): orange, blue, and green colors with lower reflectance values than those tested in 2019, a black color (commonly used in different countries [51,53]), and a pure white color, for comparison with white, unpainted coroplast. The materials and experimental design were similar to those used in 2019, so we describe below only those elements that were different. The vanes of the traps were painted twice with glossy acrylic spray paints (green from the ColorMatic series and other colors from the PrismaColor series produced by Schuller Eh‘klar, München, Germany) and then treated twice with the dry PTFE (MotipDupli Group B.V., Wolvega, The Netherlands). It is important to mention that, after spraying the vanes with the pure white paint, the final color looked rather yellowish.

The experiment was conducted in the Międzychód FD (N 52.6684, E 15.8630) and Wronki FD (N 52.7608, E 16.2576), in 65–90-year-old pure Scots pine stands with mosses in a vegetation cover. Stands were selected using the same criteria as in the first experiment. Color treatments were tested in 10 blocks, and two block conformations were used, either 3 × 2 or 6 × 1, as dictated by the terrain relief. Traps were deployed on 29 and 30 June, and then checked and emptied on 15 and 29 July. Among the insects captured, we counted only *M. galloprovincialis* (males and females separately), *S. buprestoides*, *T. formicarius*, and *T. femoralis*.

### 2.3. Statistical Analyses

The effect of color on the ratio of the number of *M. galloprovincialis* that chose either of two colors in the laboratory experiments (compared to the 50%:50% ratio) was estimated using the 𝜒^2^ test for 2 × 2 contingency tables [54].

Prior to the analyzes of field test results, data from all trap inspections in the same experiment were pooled for each trap, in order to obtain the total number of beetles. When traps in different blocks were set on two consecutive days, catches were standardized to the same exposure period by dividing the number of beetles by the actual number of exposure days and then multiplying by the greatest number of days in the period.

The effects of color (an independent variable) in the field experiments on the total number of beetles of different species captured in each experiment (dependent variable) were estimated using a generalized linear mixed model with a Conway–Maxwell–Poisson, a generalized Poisson, or a negative binomial distribution of the dependent variable. The block was considered as a random factor. The significance of the color effect was tested with a Wald 𝜒^2^ test [55]. This was followed by a Dunnett test for comparing the mean catches in the reference trap, the unpainted white trap, and the mean catches in the trap of each of the other colors tested.

Statistica 10 software [56] was used for the 𝜒^2^ test for 2 × 2 contingency tables, and all other data analyses were performed using R environment, version 4.0.3 [57], with RStudio, version 1.1.463 [58]. The following R packages were used: glmmTMB [59] for GLMM, car [60] for the Wald 𝜒^2^ test, and emmeans [61] for the Dunnett test. The goodness of fit of each model was estimated by checking for overdispersion and residual diagnostics [62,63]. The significance level was set at 𝛼 = 0.05 for all analyses.

## 3. Results

### 3.1. Response of Immature and Mature M. galloprovincialis to Colors in the Laboratory Studies

In the first laboratory test, with contrasting colors, pastel orange invoked a significantly stronger response than gentian blue in immature males (𝜒^2^ = 8.5, df = 1, *p* = 0.0035). This color was also preferred, when compared to flame red, by both immature males and females (Figure 3a). The leaf green and nut brown colors were slightly less attractive than gentian blue and flame red; therefore, they were also considered less preferred, in comparison to pastel orange. Mature beetles did not show a clear preference for the tested colors, but only two pairs of colors were compared.

In the second laboratory test, with closely related colors, a significant effect of color was found only in mature females, which preferred light pink over flame red (𝜒^2^ = 5.9, df = 1, *p* = 0.0149) (Figure 3b). This color also invoked a positive response in mature males. For other pairs of compared colors, mature beetles slightly preferred pastel yellow over pastel orange, red lilac over pastel violet (females only), cyan green over leaf green, and reseda green over fern green. In immature beetles, the largest difference in a response was between sky blue and gentian blue among the males, with the former color more preferred than the latter one (Figure 3b). Immature males showed a slight preference for reseda green over fern green, while immature females slightly preferred leaf green over cyan green. For the other colors compared, beetle responses were not clear.

In the third laboratory test, comparing a range of colors to the white, unpainted coroplast, surprisingly, cyan blue was the only color that elicited a stronger, but not significantly, response than the white color (Figure 4). The smallest difference in beetle preference, compared to the white, unpainted color, was observed for cyan blue among the blue colors, reseda green among the green colors, pastel yellow, and red lilac. The strongest and significant negative response was caused by pastel orange in both sexes of mature beetles (females—𝜒^2^ = 9.32, df = 1, *p* = 0.0023; males—𝜒^2^ = 4.6, df = 1, *p* = 0.0321) and immature females (𝜒^2^ = 5.9, df = 1, *p* = 0.0149) (Figure 4). Mature females also avoided gentian blue (𝜒^2^ = 9.3, df = 1, *p* = 0.0023), leaf green (𝜒^2^ = 4.6, df = 1, *p* = 0.0321), and fern green (𝜒^2^ = 13.9, df = 1, *p* = 0.0002). Besides, the leaf green color elicited a significantly weaker response than the white coroplast in immature males (𝜒^2^ = 7.5, df = 1, *p* = 0.0062) (Figure 4).

### 3.2. Effect of Trap Color in the Field Studies

#### 3.2.1. *M. galloprovincialis*

Based on the results of the laboratory tests, the effect of the pastel yellow, cyan blue, and reseda green colors was tested in the field in 2019, using the white, unpainted coroplast as a reference. We found no significant difference in the catches of *M. galloprovincialis* of either sex when compared to the white traps. The catches of females (40.7 ± 4.50 beetles/trap, i.e., estimated marginal mean ± SE) and beetles of both sexes (46.8 ± 4.87 beetles/trap) in the unpainted white traps were slightly higher than in the other traps (Figure 5), whereas beetles were least numerous in the cyan blue traps (28.7 ± 3.65 beetles/trap and 33.5 ± 3.97 beetles/trap, respectively). The number of males varied from 4.6 ± 0.88 beetles/trap in the cyan blue traps to 6.2 ± 1.08 beetles/trap in the pastel yellow traps (Figure 5).

The number of females in the traps was much higher than that of males, with their ratio averaging from 6.6 in the cyan blue traps to 11.5 in the unpainted white traps. All females examined had developed eggs in their abdomens. The number of eggs ranged from 13.7 ± 0.35 eggs/female/trap to 14.4 ± 0.34 eggs/female/trap (mean ± SE).

In 2020, in the experiment with the colors of lower reflection values, the trap color had the significant effect on the catches of *M. galloprovincialis* females (𝜒^2^ = 36.6, df = 5, *p* < 0.0001), males (𝜒^2^ = 15.6, df = 5, *p* = 0.0079) and total number of the pine sawyers (𝜒^2^ = 29.5, df = 5, *p* < 0.0001). The unpainted white traps captured the highest numbers of females (14.0 ± 2.32 beetles/trap), males (4.7 ± 1.11 beetles/trap), and beetles of both sexes (18.6 ± 3.08 beetles/trap). The catches of females were significantly different from those in the light blue traps, pure white traps, and pine green traps (5.3 ± 1.12, 5.6 ± 1.15, and 6.6 ± 1.30 beetles/trap, respectively) (Figure 5). The difference in the number of females in the unpainted white and jet black traps (8.9 ± 1.61 beetles/trap) was nearly significant (*p* = 0.0528). A similar pattern was observed in the total number of beetles, while the catches of *M. galloprovincialis* males were significantly lower only in the pure white and light blue traps (1.5 ± 0.46 and 1.6 ± 0.48 beetles/trap, respectively) (Figure 5).

#### 3.2.2. *S. buprestoides*

Overall, the number of *S. buprestoides* beetles captured in the traps was relatively low. The highest numbers of beetles were found in the unpainted white traps, in both 2019 and 2020 (11.4 ± 2.41 beetles/trap and 24.4 ± 3.92 beetles/trap, respectively) (Figure 5). The lowest catches were in the cyan blue traps in 2019 (5.7 ± 1.48 beetles/trap) and pure white traps in 2020 (13.5 ± 2.20 beetles/trap). The effect of trap color on the number of *S. buprestoides* beetles captured was significant only in 2020, between the unpainted white traps and painted pure white traps (Figure 5).

#### 3.2.3. *Thanasimus* spp.

In both 2019 and 2020, the catches of *T. femoralis* were much higher (approximately 25 and 4 times, respectively) than those of *T. formicarius*, regardless of trap color (Figure 5). In 2019, trap color had no significant effect on the catches of *Thanasimus* spp. Both species were slightly more numerous in the reseda green traps (*T. formicarius*—2.4 ± 0.70 beetles/trap, *T. femoralis*—59.0 ± 9.70 beetles/trap) than in the traps of the other tested colors (Figure 5). In 2020, trap color had a significant effect on the catches of both species (*T. formicarius*—𝜒^2^ = 26.5, df = 5, *p* < 0.0001, *T. femoralis*—𝜒^2^ = 37.9, df = 5, *p* < 0.0001). The highest numbers of beetles (*T. formicarius*—26.6 ± 4.90 beetles/trap, *T. femoralis*—98.8 ± 17.26 beetles/trap) were captured in the unpainted white traps and were significantly different from those in the pure white, light blue, and pine green traps (Figure 5).

## 4. Discussion

### 4.1. Response of M. galloprovincialis to Color

Numerous studies have been conducted to find the most effective traps for capturing *Monochamus* beetles, including *M. galloprovincialis*, but most of them focused on trap types [8,9,51,64], collection cups [10,65,66,67], and the effect of lubricants on their effectiveness [10,11,13,68]. They led to the development of optimal trap types, i.e., either funnel or cross-vane traps, treated with a lubricant, with some modifications of collection cups for catching live individuals [12,14]. However, much less attention has been paid to trap color. 

In general, black traps have been commonly used as the most effective traps for capturing *Monochamus* species [69,70,71] and many other bark- and wood-boring insects [72]. The black color and large size of traps seems to imitate a silhouette of host trees [8,9,16]. The first studies to test the effect of color on catches of *Monochamus* beetles were conducted in Canada, using cross-vane traps (black vs. transparent vanes [8]) and 12-funnel traps (black vs. white [16]). In the former study, the black traps captured significantly more beetles of *M. scutellatus* and *M. mutator* (LeConte) than the transparent traps, but trap color had no effect on *M. notatus*. In the latter study, the black traps were significantly more efficient than the white traps in capturing both *M. scutellatus* and *M. clamator* (LeConte). 

In Europe, a few studies on the design of traps for catching *M. galloprovincialis* have also tested the effect of color (black, white, and/or transparent) [14,15,51]; however, the results were inconclusive because of either the incomplete experimental design [14,51] or low insect catches [15]. The only study that tested more colors than those mentioned above, in traps for catching longhorned beetles, was recently conducted in Italy [17]. The efficacy of traps in one of seven colors—yellow (RAL 1018), red (RAL 3020), purple (RAL 4008), blue (RAL 5015), green (RAL 6037), gray (RAL 7034), and brown (RAL 8002)—was compared with the efficacy of black traps (RAL 9005). Overall, trap color had no significant effect on the catches of beetles from the subfamily Lamiinae, represented mainly by *Aegomorphus clavipes* (Schrank) and *Leiopus nebulosus* (L.). The catches of *M. galloprovincialis* were extremely low (1 specimen in the gray traps).

Therefore, to the best of our knowledge, our experiments are the first studies focused on testing the response of *M. galloprovincialis*, both immature and mature beetles, to a wide range of colors. A total of 20 colors were tested, of which 11 were tested only in the laboratory, 4 were tested both in the laboratory and field, and 5 were tested only in the field. The laboratory tests showed that light colors generally elicited stronger responses in the beetles than dark colors. The strongest positive response was to pastel orange vs. gentian blue and flame red in the immature beetles, as well as to pastel yellow vs. pastel orange, light pink vs. flame red, and cyan green vs. leaf green in the mature beetles. Surprisingly, none of the colors tested appeared more attractive than the unpainted white coroplast. In addition, pastel orange, gentian blue, leaf green, and fern green were significantly less attractive than unpainted white coroplast, especially for mature females. The first field test confirmed the results of the laboratory studies and showed no significant differences in the number of *M. galloprovinicialis* beetles, regardless of sex, in the traps painted with light colors (cyan blue, reseda green, and pastel yellow), compared to the unpainted white traps with the highest catches of beetles. In the second field test, the colors tested were deep orange, light blue, and pine green, which had lower reflectance than the corresponding colors in the first test, as well as jet black, pure white, and unpainted white coroplast. Unexpectedly, the catches were highest in the unpainted white traps, just as in the first test, followed by deep orange and jet black, while the pure white, light blue, and pine green traps had the lowest numbers of beetles.

Insect attraction to traps of different colors seems to depend on both color (wavelength) and brightness (reflectance) [49], and their response to color is species- and sex-specific [30,34,36,37,38]. The application of lubricants or adhesives usually increases reflectance, but has no effect on wavelength [13,34,73]. In our studies, we did not measure the reflectance of the coroplast painted using acrylic spray paints of different colors. Therefore, in further discussions, we rely on the theoretical characteristics of colors listed in Table 1. We are aware that the total reflectance intensity of different colors may vary, depending on the trap material and paint type. However, we assume that the theoretical and real wavelengths of peak reflectance are relatively close.

Our intention in testing blue, green, and orange was to use colors typical for the environment of *M. galloprovincialis*, i.e., the blue sky, green needles, and orange bark of the host tree, *Pinus sylvestris* L. Blue colors appeared unattractive to the beetles, and this may be related to the loss of blue sensitivity in many beetles, including *M. alternatus* [33]. In the green spectrum, pine green (~518 nm), which was thought to mimic pine needles and is closer to the blue spectrum, seems to be less attractive than reseda green (~562 nm), with a peak reflectance wavelength closer to the yellow spectrum and actual reflectance of *P. sylvestris* needles [74]. The positive response of *M. galloprovincialis* to green may be explained by the presence of green-sensitive opsins in the eyes of different beetles, including *M. alternatus* and *Anoplophora glabripennis* (Motschulsky) [33]. This may also be a reason why immature males and both mature males and females of *M. galloprovincialis* are attracted to the green pine plants/branches [47]. Interestingly, *M. alternatus* was found to possess only green sensitive opsins [33], but the beetles were more attracted to black and brown traps than to green traps in the field tests [46]. The difference in reflectance patterns (peak wavelengths and reflectance intensity) of tested green colors and surface types (trap material, needles, etc.) could be responsible for the different response of *Monochamus* spp., likewise the jewel beetle *A. planipennis* [34,49].

In contrast to the green colors tested, both orange colors, pastel yellow and deep orange, used to simulate the bark of *P. sylvestris* were very attractive to *M. galloprovincialis*. They were similar in wavelengths of peak reflectance (~585 and ~592 nm, respectively), but differed in reflectance intensity (Table 1). The wavelengths of their maximum reflectance were similar to those of the peak reflectance of the orange-gray part of the *P. sylvestris* bark [75] and the bark of *Pinus ponderosa* Douglas ex P. Lawson and C. Lawson, sampled at a height of 1.7–2.0 m [76], within an orange spectrum. This spectrum is a zone where the first stronger increase in bark reflectance is observed in various pines before reaching its maximum in the red spectrum of visible light [75,76,77,78].

Surprisingly, the catches of *M. galloprovincialis* in the unpainted white cross-vane traps were higher than in the jet black cross-vane traps, and the difference was nearly significant in females. This result is in contrast to those observed in *M. scutellatus* and *M. clamator* captured in the white and black funnel traps in Canada [16], and to the general trend of using black traps as more efficient for monitoring *Monochamus* species. It is well-known that white and black colors are characterized by the highest and lowest total light reflectance, respectively (see Table 1) [76,77,79], so our results suggest that *M. galloprovincialis* may, if at all, prefer colors with higher reflectance. This is supported by similar beetle catches in the traps of different, but light, colors in the first field experiment. On the other hand, the white and black colors of the unpainted coroplast and funnel traps have relatively similar and stable reflectance over almost the entire range of the visible spectrum, starting at about 420 nm, with some small peaks that often coincide at the same wavelengths [76,80]. This aspect was specifically addressed in the study of spectral reflectance of unpainted coroplast of different colors and other materials (Lindgren funnel trap and bark of *Pinus radiata* D. Don), when the relative reflectance for each color was standardized by using a reflected peak wavelength of that color to eliminate differences in total reflectance between colors [77]. Both the white and black coroplast and black funnel trap had the highest relative intensity of sunlight reflectance at the same wavelengths, approximately 500 nm, which corresponds to sunlight reflectance, and 540–550 nm. The greater importance of similarity of reflectance patterns than reflectance intensity for the attractiveness of different colors to some insects is supported by the lack of differences in the catches of stable flies (Diptera, Muscidae) in white and gray traps, with similar reflectance patterns over a wide range of the visible spectrum, while the reflectance intensity of the former traps was more than 2.5 times higher than that of the latter traps [80].

Another interesting and unexpected finding of our studies was the lowest effectiveness of the pure white traps. They were expected to catch as many *M. galloprovincialis* as the unpainted white traps because of their high reflectance intensity (Table 1). However, these traps differed in their actual colors. The pure white color appeared yellowish on the traps and seems to correspond to the wavelength (~576 nm) indicated in Table 1. Therefore, the differences in the catches of *M. galloprovincialis* between the unpainted white traps and painted pure white traps could be explained by the differences in reflectance patterns, rather than reflectance intensity, as already highlighted above.

A stronger response of *M. galloprovincialis* to black and orange traps could also be related to the colors of the beetle elytra, likewise in *A. planipennis* [34]. Orange and black on beetle elytra may be considered as an evolved ability to mimic the insect’s environment and thus increase survival while feeding and searching for oviposition sites (as has been shown in some chrysomelids [27]), with the black color presumably useful for hiding in shady places and when staying on a dark background. If *M. galloprovincialis* is considered a diurnal insect, elytra colors are likely used as a visual cue for mate finding, likewise in other cerambycids, e.g., the black color preferred by *Anoplophora malasiaca* (Thomson) [23,24] and dark-blue preferred by *Zorion guttigerum* Westwood [81], as well as in other insects, e.g., *Agrilus* spp. [26,34,36].

In both field experiments, the catches of *M. galloprovincialis* males were much lower than those of the females, that was expected because of the lure composition, consisting of an aggregation pheromone released by males, but also semiochemicals released by host trees and bark beetles, which inform *M. galloprovincialis* females about suitable breeding sites [82]. Analyzes of the captured females showed that all were sexually mature, confirming that the Galloprotect Pack lure attracts only mature individuals [83].

### 4.2. Response of Non-Target Insect Species to Color

The most abundant non-target species captured in the traps in our studies were *S. buprestoides* and two species of clerid beetles: *T. femoralis* and *T. formicarius*. Our results confirm those obtained by other researchers in Europe [17,71,84,85,86].

The catches of *S. buprestoides* were only weakly dependent on trap color, with the highest number of beetles captured in the unpainted white traps. In our earlier tests of trap type and color (black, white and transparent), the catches of these beetles were also affected by trap type rather than trap color [15]. *S. buprestoides* was most abundant in the black 12-funnel traps, while the lowest catches (10-fold difference) were in the cross-vane traps (white and transparent). These results were the basis for choosing cross-vane traps in further studies. Interestingly, the effect of trap type was significant only when the lure composition contained ethanol and α-pinene. The significant effect of both chemicals or ethanol alone (when added to the tested lures) on *S. buprestoides* was shown in a few studies [71,84,87,88]. The attraction of this insect species to ethanol-containing lures results from the biology of *S. buprestoides*, i.e., its development in conifer stumps, which produce a relatively high ethanol concentration [89].

The effect of trap color on the catches of *T. formicarius* and *T. femoralis* was similar and followed the pattern found for the total catches of *M. galloprovincialis*. In the first field test, no significant differences were found between the unpainted white traps and the other trap colors, but the highest numbers of *T. femoralis* were captured in the reseda green traps. In the second field test, both species were most abundant in the unpainted white traps, followed by the jet black and deep orange traps, while significantly fewer beetles were captured in the pure white, light blue, and pine green traps. When comparing our results to those of other studies, much attention should be paid to the types of traps used by other researchers and whether or not the traps were painted. Our studies revealed that painting the traps with pure white paint (according to RAL) results in a yellowish coloration of the surface, which significantly reduced the trap attractiveness to the insects studied. Painting 16-funnel traps, as well as *Pinus taeda* L. trees, with white or black paint resulted in a significant decrease in both the number of bark beetle attacks on trees and number of the southern pine beetle, *Dendroctonus frontalis* Zimmermann, and *Thanasimus dubius* (F.) beetles in the white traps [90]. When unpainted sticky traps made of clear, white, and black Plexiglas were tested, the black traps were also more attractive to *T. dubius* than white traps and transparent traps [90]. In our previous studies testing unpainted traps of different types and colors, the catches of *T. formicarius* had an increasing trend from transparent cross-vane traps through white cross-vane and six-funnel traps to black six-funnel traps, while the numbers of *T. femoralis* were comparable among these traps types [15]. When more colors were included in the tests, black, brown, gray, and purple appeared to be more attractive to *T. formicarius* than blue, green, and yellow in the cross-vane painted traps [86]. In other studies, green three-funnel traps captured as many *T. formicarius* as the black traps, and white and yellow traps were the least attractive [91]. The difference in the attractiveness of green traps in the two studies mentioned above and in our studies, as well as between unpainted and painted white traps, suggests that the reflectance spectrum of color is important not only for *M. galloprovincialis*, as discussed earlier, but also for *T. femoralis* and *T. formicarius*. 

The preference of *T. formicarius* and *T. femoralis* for the unpainted white, black, orange, and green traps in our studies could be due to their possible role in mate finding (black, white, and orange-red colors of the beetles), but also in finding the host trees of their preys, however further studies are needed to prove this hypothesis.

Surprisingly, we observed a high *T. femoralis* bias in the number of clerid beetles captured, regardless of trap color. In general, one might expect much higher numbers of *T. formicarius* in Scots pine stands (with rather low numbers of *I. typographus*, if any) and when Galloprotect Pack lure, with the chemicals more attractive to *T. formicarius*, is used [92,93,94,95]. The only reasonable explanation for the bias of *T. femoralis* could be related to the differences in the flight periods of these species. *T. formicarius* starts flying much earlier than *T. femoralis*, so its abundance decreases before the end of June, whereas *T. femoralis* seems to be abundant until late July–late August [85,93,95,96,97].

## 5. Conclusions

The results of our studies, and a wide range of available data [18,27,36,37,38,73,86,98,99,100,101] and references therein, indicate that the insect response to different colors is species- and sex-specific and depends largely on their host plants, but also on the coloration of the insects themselves (elytra and body). Our laboratory tests suggest that immature and mature beetles of *M. galloprovincialis* differ in their color preference; however, this finding should be verified in further studies. Of the trap colors tested against the mature beetles in the field experiments, the unpainted white traps were found to be most efficient. However, green with a peak reflectance wavelength closer to the yellow spectrum, orange (both colors tested), and black (especially for males) were also very attractive to the mature beetles and are likely visual cues for *M. galloprovincialis* when searching for host trees as food sources and oviposition sites and/or for mates. A similar pattern of response to color was observed in the predatory clerid beetles *T. femoralis* and *T. formicarius*. The spectral composition of light reflectance from traps seems to be more important than reflectance intensity in the response of these three insect species to color. In contrast, trap color appeared to be less important than trap type for *S. buprestoides*, particularly when ethanol was included in a lure. The similar responses of *M. galloprovincialis* and two clerid species to the trap colors tested make optimization of traps by selecting a color specific to *M. galloprovincialis* rather impossible. Therefore, either the traps should be modified to allow insects smaller than *M. galloprovincialis* to leave the insect collectors [102], or the lures should be modified to make them less attractive to non-target species, especially beneficial insects.

## Figures and Tables

**Figure 1 insects-13-00220-f001:**
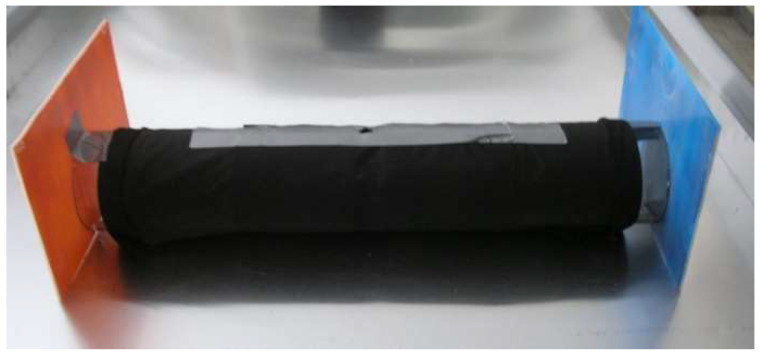
The experimental design for testing the response of *M. galloprovincialis* to color under laboratory conditions.

**Figure 2 insects-13-00220-f002:**
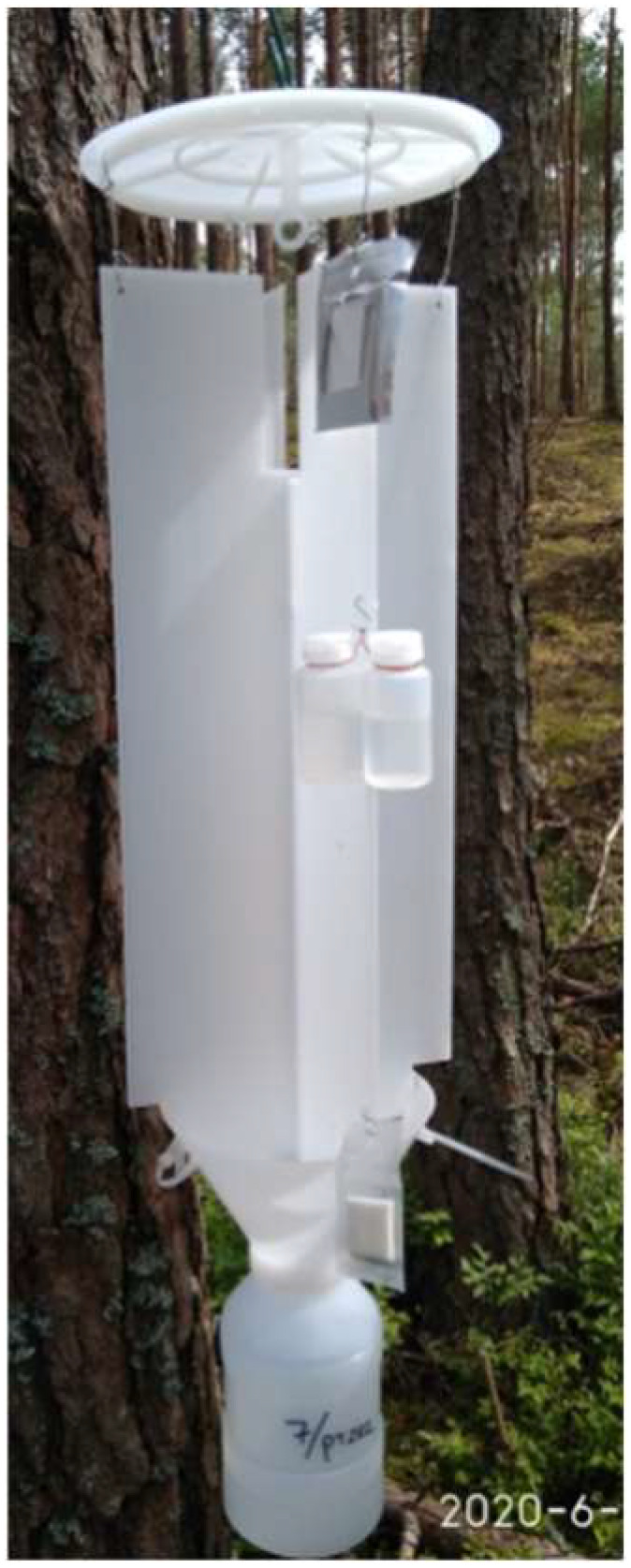
The unpainted, white cross-vane trap IBL-5, with dispensers located according to the instruction provided by the producing company SEDQ (Spain): the dispenser with bark beetle kairomones is located in the upper part of the trap, two dispensers with α-pinene in the middle, and the dispenser with the aggregation pheromone in the lower part of the trap.

**Figure 3 insects-13-00220-f003:**
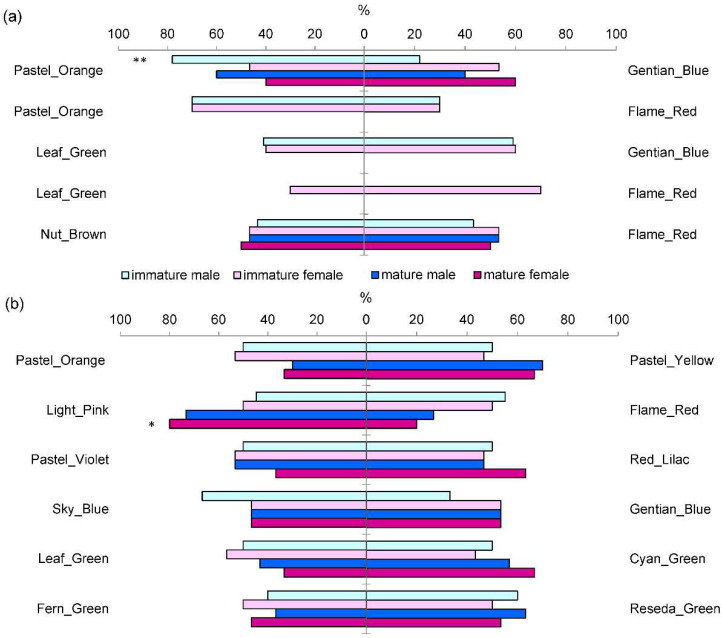
Percentage of immature and mature *M. galloprovincialis* beetles that responded to one of the compared pairs of (**a**) contrasting colors and (**b**) closely related colors under laboratory conditions; *, ** indicate significant differences from the 50%:50% ratio with *p* < 0.05 and *p* < 0.01, respectively (tested with the 𝜒^2^ test for 2 × 2 contingency tables using numbers of beetles provided in Appendix A).

**Figure 4 insects-13-00220-f004:**
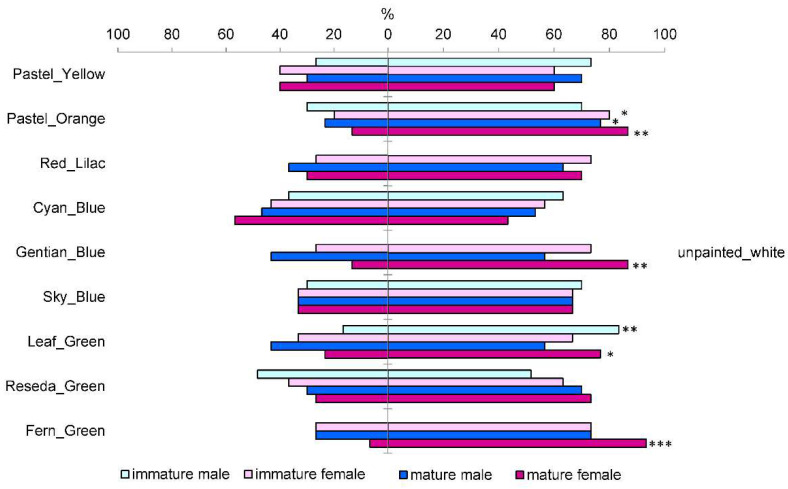
Percentage of immature and mature *M. galloprovincialis* beetles that responded to either the tested color or the unpainted white (coroplast) under laboratory conditions; *, **, *** indicate significant differences from the 50%:50% ratio with *p* < 0.05, *p* < 0.01, and *p* < 0.001, respectively (tested with the 𝜒^2^ test for 2 × 2 contingency tables using numbers of beetles provided in Appendix A).

**Figure 5 insects-13-00220-f005:**
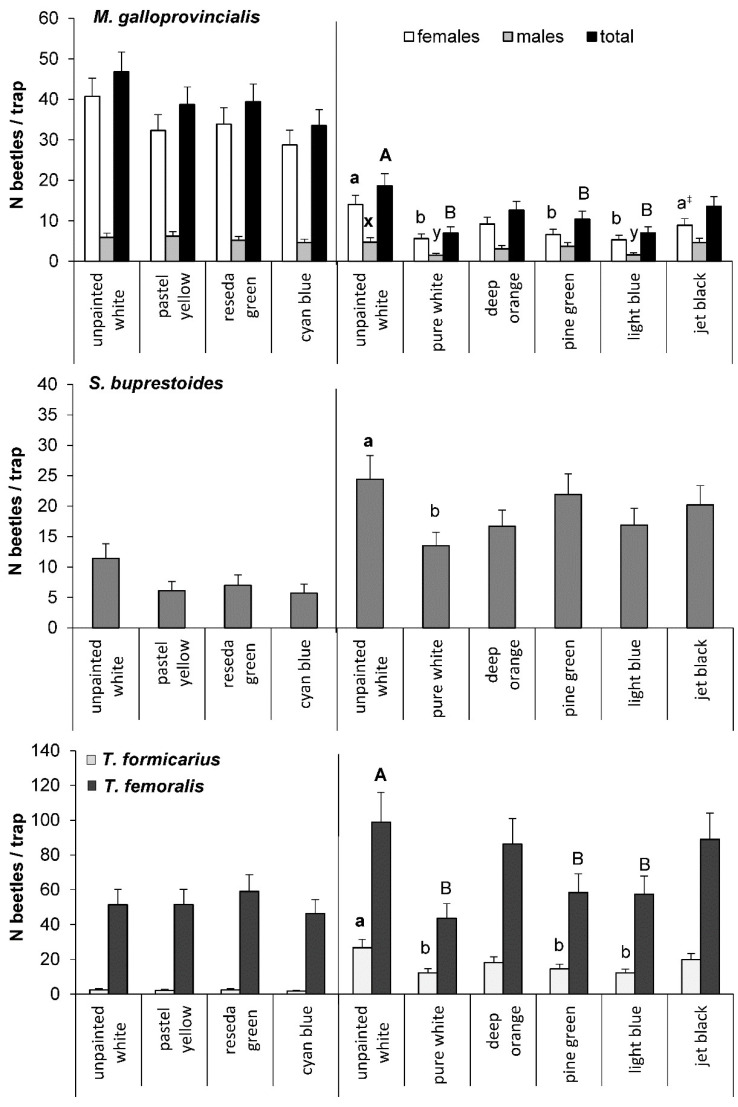
The number (estimated mean and SE) of beetles of *M. galloprovincialis* and three most abundant insect species captured in the traps of different colors: 2019, 24 June–9 July for *M. galloprovinicialis*, 24 June–1 July for other species (N = 14 traps/color); 2020, 29 June–29 July (N = 10 traps/color); different letters above bars indicate significant differences between the traps of different colors and the unpainted white (reference) traps within sex or species for each year separately at *α* = 0.05, ^‡^ the difference was nearly significant (*p* = 0.0528). Raw numbers of captured beetles are provided in Appendix A.

**Table 1 insects-13-00220-t001:** A list of colors tested in the laboratory and/or field studies.

Color	Codes According to RAL or NCS ^1^ Color Systems	Approx. Wave- Length ^2^	Light Reflectance Value ^2^	Laboratory Tests			Field Studies	
				1	2	3	2019	2020
Pastel Yellow	1034	585.4	42.1		x ^3^	x	x	
Pastel Orange	2003	592.7	33.2	x	x			
Deep Orange	2011	591.7	27.4					x
Flame Red	3000	608.0	9.9	x	x			
Light Pink	3015	669.7	42.5		x			
Red Lilac	4001	551.5	14.9		x	x		
Pastel Violet	4009	510.0	26.3		x	x		
Cyan Blue	S 0520-R90B ^1^	481.8	73.0		x	x	x	
Gentian Blue	5010	478.3	7	x	x	x		
Light Blue	5012	481.9	21.3					x
Sky Blue	5015	480.4	17.5		x	x		
Cyan Green	S 4030-B90G ^1^	506.5	14.4		x			
Leaf Green	6002	553.5	8		x	x		
Reseda Green	6011	562.3	18.3		x	x	x	
Fern Green	6025	564.5	13.8		x	x		
Pine Green	6028	518.4	7.4					x
Nut Brown	8011	589.4	5.1	x				
Jet Black	9005	464.2	0.5					x
Pure White	9010	575.7	83–96					x
White (unpainted) coroplast						x	x	x

^1^ NCS—Natural Color System^®^, ^2^ according to Encycolorpedia [52], ^3^ color included in the laboratory tests/field studies.

**Table 2 insects-13-00220-t002:** Design of the laboratory tests and numbers of responding *M. galloprovincialis* beetles (non-responding individuals were excluded from the sample size).

Test	Pair of Tested Colors	Immature Beetles		Mature Beetles	
		Males	Females	Males	Females
1	Pairs of contrasting colors				
	Pastel Orange/Gentian Blue	50	88	30	30
	Pastel Orange/Flame Red	20	20	-	-
	Leaf Green/Gentian Blue	22	20	-	-
	Leaf Green/Flame Red	-	20	-	-
	Nut Brown/Flame Red	30	30	30	30
2	Pairs of closely related colors				
	Pastel Orange/Pastel Yellow	30	30	30	30
	Light Pink/Flame Red	29	30	30	30
	Pastel Violet/Red Lilac	30	30	30	30
	Sky Blue/Gentian Blue	30	30	30	30
	Leaf Green/Cyan green	30	30	30	30
	Fern Green/Reseda Green	30	30	30	30
3	Colors tested against white (unpainted) coroplast				
	Pastel Yellow	30	30	30	30
	Pastel Orange	30	30	30	30
	Red Lilac	-	30	30	30
	Cyan Blue	30	30	30	30
	Gentian Blue	-	30	30	30
	Sky Blue	30	30	30	30
	Leaf Green	30	30	30	30
	Reseda Green	29	30	30	30
	Fern Green	-	30	30	30

## Data Availability

The raw data are available in Appendix A.

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
