# Peer review of "The Effect of Trap Color on Catches of Monochamus galloprovincialis and Three Most Numerous Non-Target Insect Species"

_insects, 2022, doi:10.3390/insects13030220_

Round 1

Reviewer 1 Report

Several methods of sampling forest insects do exist. The trapping programs for species of agricultural importance use a variety of attractant cues (chemical and visual). Species-specific sampling protocols use attractant-baited traps (pheromones, kairomones) to lure insects to traps. Different types of traps concerning design and structure are employed. Based on the insect sensitivity to different colors, entomologists use passive and active traps to collect insects. However, trap color is barely considered in the entomological studies of Monochamus spp. This is exactly what this research does by investigating the effect of trap color on the response of the pine sawyer Monochamus galloprovinciallis. This study is an interesting field of work and present good results, but some issues deserve further comments.

In the introduction, it would be interesting to mention that, besides color and shape, trap position in terms of height above ground level also affects the catch size.

It was assumed that color traps would perform better than the conventional black traps. However, in the introduction, no complete information is presented regarding the role of vision of these insects in navigation (wavelength range). Can beetles distinguish trap colors and reflected light from different surfaces (habitats/hosts)?

What is known about the photoreceptors that detect different wavelengths of light in immature beetles?

Lines 116-117: the same as in the lines 76-77?

Line 130: what components make up the Galloprotec pack?

Line 132 (figure 2): indicate with an arrow or words the position of the chemical attractants, specifying the type of lure employed (put in the figure caption).

Lines 291-337: unnecessary to repeat these arguments here at length.

Line 369: first appearance; must be written in full – Pinus sylvestris.

Author Response

Response to Reviewer 1

In our response to each Reviewer, we start from general information about the changes made in our manuscript and this information is the same in each response. Our responses to specific comments are provided further, in the part Comments and Suggestions for Authors.

First of all, we would like to thank for all comments and questions. We did our best to meet the reviewer expectations. However, sometime we felt that some issues lay beyond the topic of our manuscript and in such cases we did not elaborate/include those aspects in the manuscript.

In addition to the changes made in the text in response to the reviewers’ comments, we also corrected some mistakes/spells/style we found on our own. All of them were done in Track changes mode. Substantial changes were done in Abstract, because we wanted to highlight that colors were tested also against immature beetles, while this aspect was overlooked in the original version of the manuscript. Of the changes made in the manuscript body, I would like to explain those in lines 264-265 (the revised version with visible changes). In the original version of the manuscript, the results of chi square test were assigned to the wrong sexes and we detected it while preparing a new version of Figure 4. We corrected it in the revised manuscript.

While adding new information, e.g., in the introduction, we had to add new references, thus the numbering of citations has changed. To keep the text clearer, we allowed ourselves to do these changes, i.e., numbering of citations and new references in Reference, without tracking these changes.

Line numbering we used below is related to the version with visible changes.

Comments and Suggestions for Authors

Several methods of sampling forest insects do exist. The trapping programs for species of agricultural importance use a variety of attractant cues (chemical and visual). Species-specific sampling protocols use attractant-baited traps (pheromones, kairomones) to lure insects to traps. Different types of traps concerning design and structure are employed. Based on the insect sensitivity to different colors, entomologists use passive and active traps to collect insects. However, trap color is barely considered in the entomological studies of Monochamus spp. This is exactly what this research does by investigating the effect of trap color on the response of the pine sawyer Monochamus galloprovinciallis. This study is an interesting field of work and present good results, but some issues deserve further comments.

Comment 1 – In the introduction, it would be interesting to mention that, besides color and shape, trap position in terms of height above ground level also affects the catch size.

ResponseIn our manuscript we focused just on a trap itself. Trap location is a completely different topic, with many different aspects, e.g. height (crown – ground), site type (from open field to closed canopy), trap position in relation to the trapping area (western-eastern or other), etc. Thus, we have decided to not “touch” this topic in this manuscript, but we are preparing the next manuscript, in which trap location is the main topic.

Comment 2 – It was assumed that color traps would perform better than the conventional black traps. However, in the introduction, no complete information is presented regarding the role of vision of these insects in navigation (wavelength range). Can beetles distinguish trap colors and reflected light from different surfaces (habitats/hosts)?

ResponseWe are not sure whether the above questions are related to M. galloprovincialis, Monochamus spp., longhorned beetles or beetles in general? We are not aware about any detailed studies of
M. galloprovincialis eyes, their structure and sensitivity to different wavelengths, however in Introduction we added some information about color vision in insects and particularly in long-horned beetles and provided the examples available for Monochamus spp.

More complex studies on the sensitivity to different wavelengths and reflectance intensity from different surfaces were done for Agrilus planipennis and other jewel beetles and we referred to those studies in lines 426-429. We also refer to this aspect when discussing the difference/lack of difference between the catches of M. galloprovincialis to the black and unpainted white traps (lines 439-462). In our studies, we focused rather on the effect of color itself, used on the same type of surface(coroplast), to check whether some colors are more attractive than others and whether black is indeed optimal. Our results provide a kind of basis for further, more specific studies on M. galloprovincialis, and it is rather certain that they will be done in the nearest future.

Comment 3 – What is known about the photoreceptors that detect different wavelengths of light in immature beetles?

ResponseWe were not able to find any information about vision in immature beetles for Monochamus galloprovincialis, Monochamus spp. and even Cerambycidae, but we found a very interesting recently published paper about Agrilus planipennis and used it in our manuscript (lines 77-81).

Comment 4 – Lines 116-117: the same as in the lines 76-77?

Response – Corrected (lines 149-150)

Comment 5 – Line 130: what components make up the Galloprotec pack?

Response – Added (lines 163-166)

Comment 6 – Line 132 (figure 2): indicate with an arrow or words the position of the chemical attractants, specifying the type of lure employed (put in the figure caption).

Response – Added (lines 168-171)

Comment 7 – Lines 291-337: unnecessary to repeat these arguments here at length.

Response – We have decided to leave the text in lines 291-306 (current lines 335-351) almost unchanged, because in comparison with the introduction, we provide more details on the papers/authors dealing with different aspects of trap design (lines 291-296, currently lines 335-341) and also more information on the studies that supposedly leaded to the common use of black traps in monitoring of Monochamus species (lines 297-306, currently lines 342-351). For the same reason, i.e., providing more details on tested colors as a background for our studies, we left almost unchanged the text in lines 329-337 (currently lines 376-383). However, we have shortened much the part of the text, i.e. in lines 307-328 (currently lines 352-376).

Comment 8 – Line 369: first appearance; must be written in full – Pinus sylvestris.

Response – Corrected (line 415).

Reviewer 2 Report

The manuscript provides clear data that unpainted white cross-vane traps provide the best color for trapping Monochamus galloprovincialis and other non-target species. The data also shows color preferences in lab assays that were not reflected in the trap captures. The authors for the most part do a very good job of explaining the experiment and discussing the results. The following suggestions are made.

1) The presentation of Figure 3 is confusing. It would be easier to follow if another column were made on the right that described the second color used, rather than placing  the first and second colors together on the left. It would also be a better presentation to show percentages of insects going toward either color rather than the ratio. Using the ratio exagerates the preferences to the right, and minimizes the preferences to the left.

2) The discussion is quite long and could be condensed in places without providing quite as many details about the studies discussed. One possible hypothesis that is not discussed is that color may be effecting volatile emission from the lures by either photoisomerization or heat production if UV or IR wavelengths are reflected toward the lures.

Author Response

Response to Reviewer 2

In our response to each Reviewer, we start from general information about the changes made in our manuscript and this information is the same in each response. Our responses to specific comments are provided further, in the part Comments and Suggestions for Authors.

First of all, we would like to thank for all comments and questions. We did our best to meet the reviewer expectations. However, sometime we felt that some issues lay beyond the topic of our manuscript and in such cases we did not elaborate/include those aspects in the manuscript.

In addition to the changes made in the text in response to the reviewers’ comments, we also corrected some mistakes/spells/style we found on our own. All of them were done in Track changes mode. Substantial changes were done in Abstract, because we wanted to highlight that colors were tested also against immature beetles, while this aspect was overlooked in the original version of the manuscript. Of the changes made in the manuscript body, I would like to explain those in lines 264-265 (the revised version with visible changes). In the original version of the manuscript, the results of chi square test were assigned to the wrong sexes and we detected it while preparing a new version of Figure 4. We corrected it in the revised manuscript.

While adding new information, e.g., in the introduction, we had to add new references, thus the numbering of citations has changed. To keep the text clearer, we allowed ourselves to do these changes, i.e., numbering of citations and new references in Reference, without tracking these changes.

Line numbering we used below is related to the version with visible changes.

Comments and Suggestions for Authors

The manuscript provides clear data that unpainted white cross-vane traps provide the best color for trapping Monochamus galloprovincialis and other non-target species. The data also shows color preferences in lab assays that were not reflected in the trap captures. The authors for the most part do a very good job of explaining the experiment and discussing the results. The following suggestions are made.

Comment 1 – The presentation of Figure 3 is confusing. It would be easier to follow if another column were made on the right that described the second color used, rather than placing  the first and second colors together on the left. It would also be a better presentation to show percentages of insects going toward either color rather than the ratio. Using the ratio exagerates the preferences to the right, and minimizes the preferences to the left.

Response – We changed Figures 3 and 4, as suggested, and corrected captions.

Comment 2 – The discussion is quite long and could be condensed in places without providing quite as many details about the studies discussed.

Response – We have shortened the discussion at the beginning (lines 352-376), by eliminating some details. In other parts, we tried to compare our results with those obtained by other researchers and to explain various aspects of potential effect of color on the response of M. galloprovincialis. Readers interested in different species/insect behavior/color vision might gain from presentations of more details, so that everyone can find something useful.

Comment 3 – One possible hypothesis that is not discussed is that color may be effecting volatile emission from the lures by either photoisomerization or heat production if UV or IR wavelengths are reflected toward the lures.

Response – Well, it is an interesting point, but it goes far beyond the scope of our studies and would require detailed knowledge on dispensers types, temperature dependent release-rates of each, characteristics of lure components, characteristics of coroplast and colors, etc. Therefore, we decided to not include it into this manuscript.

Reviewer 3 Report

I found this a most interesting paper that was well written, Experimental methods were clearly and succinctly described. Results were appropriately analysed and displayed. 

I have only minor grammatical corrections that relate to American spelling eg color/colour. 

Section 2.2 should also read 2.2 Testing the Response of Immature and Mature M. galloprovincialis to Different colours under Field Conditions.

Line 16: 9 should be nine

Line 43: Adults feed on shoots and thin bark of twigs until or before sexual......

Author Response

Response to Reviewer 3

In our response to each Reviewer, we start from general information about the changes made in our manuscript and this information is the same in each response. Our responses to specific comments are provided further, in the part Comments and Suggestions for Authors.

First of all, we would like to thank for all comments and questions. We did our best to meet the reviewer expectations. However, sometime we felt that some issues lay beyond the topic of our manuscript and in such cases we did not elaborate/include those aspects in the manuscript.

In addition to the changes made in the text in response to the reviewers’ comments, we also corrected some mistakes/spells/style we found on our own. All of them were done in Track changes mode. Substantial changes were done in Abstract, because we wanted to highlight that colors were tested also against immature beetles, while this aspect was overlooked in the original version of the manuscript. Of the changes made in the manuscript body, I would like to explain those in lines 264-265 (the revised version with visible changes). In the original version of the manuscript, the results of chi square test were assigned to the wrong sexes and we detected it while preparing a new version of Figure 4. We corrected it in the revised manuscript.

While adding new information, e.g., in the introduction, we had to add new references, thus the numbering of citations has changed. To keep the text clearer, we allowed ourselves to do these changes, i.e., numbering of citations and new references in Reference, without tracking these changes.

Line numbering we used below is related to the version with visible changes.

Comments and Suggestions for Authors

Comment 1 – I have only minor grammatical corrections that relate to American spelling eg color/colour.

Response – We chose American English for our manuscript and used appropriate spellings for color and other words throughout the manuscript.

Comment 2 – Section 2.2 should also read 2.2 Testing the Response of Immature and Mature M. galloprovincialis to Different colours under Field Conditions.

Response – Corrected as follows: “Testing the Response of Mature M. galloprovincialis to Different Colors under Field Conditions” (lines 149-150)

Comment 3 – Line 16: 9 should be nine

Response – Corrected (line 16)

Comment 4 – Line 43: Adults feed on shoots and thin bark of twigs until or before sexual......

Response – We changed it as follows: “To reach sexual maturity, adults feed on shoots and thin bark of twigs.” (lines 46-47)